# Mass Spectrometry Study about In Vitro and In Vivo Reaction between Metformin and Glucose: A Preliminary Investigation on Alternative Biological Behavior

**DOI:** 10.3390/ijms25010180

**Published:** 2023-12-22

**Authors:** Gianluca Bartolucci, Marco Pallecchi, Laura Braconi, Silvia Dei, Elisabetta Teodori, Annunziata Lapolla, Giovanni Sartore, Pietro Traldi

**Affiliations:** 1Dipartimento di Neuroscienze, Psicologia, Area del Farmaco e Salute del Bambino (NEUROFARBA), Università di Firenze, 50100 Firenze, Italy; marco.pallecchi@unifi.it (M.P.); laura.braconi@unifi.it (L.B.); silvia.dei@unifi.it (S.D.); elisabetta.teodori@unifi.it (E.T.); 2Dipartimento di Medicina, Università di Padova, 35100 Padova, Italy; annunziata.lapolla@unipd.it (A.L.); g.sartore@unipd.it (G.S.); 3Istituto di Ricerca Pediatrica Città della Speranza, 35100 Padova, Italy

**Keywords:** metformin, reaction of metformin with glucose, mass spectrometry

## Abstract

Metformin is the most prescribed glucose-lowering drug worldwide; globally, over 100 million patients are prescribed this drug annually. Some different action mechanisms have been proposed for this drug, but, surprisingly, no metabolite of metformin has ever been described. It was considered interesting to investigate the possible reaction of metformin with glucose following the Maillard reaction pattern. The reaction was first performed in in vitro conditions, showing the formation of two adducts that originated by the condensation of the two molecular species with the losses of one or two water molecules. Their structures were investigated by liquid chromatography coupled with mass spectrometry (HPLC-MS), tandem mass spectrometry (MS/MS) and accurate mass measurements (HRMS). The species originated via the reaction of glucose and metformin and were called *metformose* and *dehydrometformose*, and some structural hypotheses were conducted. It is worth to emphasize that they were detected in urine samples from a diabetic patient treated with metformin and consequently they must be considered metabolites of the drug, which has never been identified before now. The glucose-related substructure of these compounds could reflect an improved transfer across cell membranes and, consequently, new hypotheses could be made about the biological targets of metformin.

## 1. Introduction

In the chemistry of natural products, it has been found that *Galega officinalis*, a traditional herbal medicine employed in Europe from the Middle Ages, exhibits many different medical properties mainly related to the presence of guanidine, which, in 1918, was shown to lower blood glucose levels [1]. Consequently, some studies were carried out into the synthesis of guanidine derivatives, and one of them, metformin (dimethylbiguanide, Figure 1), was firstly employed in the 1940s as antimalarial agent, but clinical tests proved that it sometimes led to a lowering of blood glucose levels. 

This property, at first sight considered negative, was studied in 1957 for diabetes treatment [2] and, due to the positive results thus obtained, metformin was introduced into clinical practice [3]. 

It is important to emphasize that metformin does not have a single mechanistic target: in fact, besides its effect of lowering glucose, it also acts in metabolic, vascular and other physiological functions [4]. However, as matter of fact, metformin is actually the most prescribed glucose-lowering drug worldwide, and over 100 million of diabetic patients are treated annually with this drug. 

But what is the action of metformin? Recently, Rena et al. [5] gave an exhaustive panorama of the actions of metformin at the molecular level described in the literature. It was proposed that metformin leads to a decreased hepatic gluconeogenesis, caused on the one hand by the modulation of mitochondrial enzymes and on the other by the regulation of the glucagon signaling pathway [6,7]. Recently, genetic loss-of-function studies indicated that 5′AMP-activated protein kinase (AMPK) participates in the anti-hyperglycemic action of metformin. As shown in Figure 2, the AMPK-independent effects of the drug include mitochondrial actions, which are considered the primary site of action of metformin.

In addition to the effect of reducing glucose levels, other beneficial effects were described, including a reduction in cardiovascular disease and mortality [8]. A possible reduction in cancer incidence was observed in some studies [9,10,11] but not all [12]. 

Organic cation transporter OCT1 is considered responsible for the transport of metformin into hepatocytes, resulting in the inhibition of the mitochondrial respiratory chain (complex I), but this is achieved through a currently unknown mechanism. The deficit in energy production so generated is balanced by reducing gluconeogenesis in the liver. It must be stressed that this hypothesis is not well related to the chemical parameters of metformin [13,14], which indicate its low lipophilicity. Consequently, the rapid passive diffusion of metformin through cell membranes seems to be considered unfavored. However, OCT1 and OCT3 have been suggested as possible transporters of metformin in the liver [15].

Surprisingly, no metabolite of metformin has ever been found or described [16,17,18,19]. By the intravenous administration of ^14^C-labelled metformin, a 100% recovery of the unchanged drug in urine was found [19]. Chromatographic assays of the unlabeled drug could not account for approximately 20% of the drug [17,18,19]. These results suggest that small proportions of doses of metformin may be metabolized or excreted by non-renal routes, even when considering that the renal clearance (CLR) of metformin is very high and it is the major mode of elimination of the drug [20]. 

A recent paper on the absorption and disposition pharmacokinetics of metformin in nine species reports that the blood-glucose-lowering effect of metformin can be attributed to several tissues [21]. In vitro studies in rodents have shown that metformin improves the insulin sensitivity of muscle cells, leading to an increased insulin-dependent glucose uptake in cells [22,23,24,25]. In addition, an increased glucose uptake into the intestinal mucosa and a reduced intestinal glucose absorption were observed in metformin-treated rats. This absorption of metformin accounts for its insignificant metabolism in the liver [21]. 

At first sight, the impressive safety, effectiveness and low cost of metformin itself might indicate that drugs that act in a similar manner are not of interest. However, some studies have reported on this topic [26].

Considering the high polarity and the strong reactivity of metformin, a question may arise weather is the original structure the active form at the cellular level, or are other related compounds, resulting from its reaction with some circulating molecular species, responsible for either pharmacokinetic and/or pharmacodynamics characteristics. In this context, it can be considered that metformin can react with circulating glucose. In fact, the reactivity between glucose and the amino groups of the lysine residues present in protein chains is well described by the Maillard reaction pattern [27]. As discussed by Gokhale et al. [28], these reactions are important for evaluating both the drug’s stability and the synthesis of related compounds. Taking into account that the formation of glycosylamines in solution, the first step of the Maillard reaction, involves a decrease in the concentration of the drug, this is significant from the point of view of its activity. On the other hand, related compounds can be active, thus modulating the pharmacokinetics and/or pharmacodynamics of the drug. 

In this paper, we report an investigation into the chemical reaction between glucose and metformin, showing the formation of new products and characterizing them using mass spectrometric techniques. It is important to emphasize that the glucose concentration in blood is in the range 3.9–5.5 mM in normal subjects, increasing up to values higher than 7.5 mM for diabetic patients. For a 1000 mg assumption, the metformin Cmax is 0.015 mM, while in the steady state it is in the range of 0.002–0.011 mM. Then, the possible reaction of metformin with circulating glucose is to be considered. As a matter of fact, in this investigation, the products originating from the glucose–metformin reaction were found in urine samples of diabetic subjects treated with metformin, suggesting their active role in metformin’s metabolism.

## 2. Results and Discussion

In previous experiments, we studied the reaction of sugars with proteins, particularly the interactions of glucose with the ε-amino groups of the lysine residues belonging to the protein chains [29]. These reactions follow the Maillard reaction pathway [27], i.e., the mechanisms of the reaction between sugars and amino groups. Then, the possible reactions between metformin and circulating glucose would be, in principle, highly favored.

### 2.1. Metformin–Glucose Reaction

The first step of the Maillard reaction is a sugar–amine condensation reaction that forms an N-substituted glycosylamine [30]. The second step in the Maillard reaction is the Amadori rearrangement, an isomerisation reaction that results in the formation of a ketosamine, called the Amadori compound, containing both a ketose (a sugar bearing a ketone) and an amine (Figure 1).

If we apply the Maillard mechanism to the reaction between glucose and metformin, the pathway reported in Figure 2 is obtained.

Since the Amadori compound was obtained from the reaction between glucose and metformin, to facilitate its identification in the text, it is called *metformose*. It can be observed that it maintains the main chemical properties of both glucose and metformin, but it surely exhibits a bioactivity that is necessarily different to that of metformin. Does this reaction occur in the experimental conditions reported in the experimental section?

### 2.2. MS Analysis of the Glucose/Metformin Reaction Mixture

As preliminary approach, we investigated the composition of the glucose/metformin reaction mixture (working solution 1), obtained as described in Section 3.2, by an MS study including ESI-positive ion mass spectra acquisitions (MS), accurate mass measurements (HRMS) and collisional induced decompositions (MS/MS). The ESI-MS spectrum of positive ions thus obtained is reported in Figure 3.

Considering that this mass spectrum was obtained by using an electrospray ionization (ESI) source, working solution 1 was most likely composed of a mixture of compounds, highlighting the incomplete purification of the raw reaction mixture during the flash chromatography (see Section 3.2). However, three ion signals caught our attention, i.e., those at *m*/*z* 130, 274 and 292, which should represent molecular structures of interest. The cluster signal at *m*/*z* 130 was assigned to the protonated ion of unreacted metformin still present in the analyzed solution, although in a low quantity. The highest ion signal at *m*/*z* 274, along with that at *m*/*z* 292, represented the compounds obtained by the reaction between glucose and metformin. Indeed, the ion at *m*/*z* 292 could be the protonated form of *metformose*, i.e., the structure proposed in Figure 2, while that at *m*/*z* 274 could have originated from metformose via water loss (*dehydrometformose* or *DHmetformose*). The conversion of *metformose* in *DHmetformose* could happen both in the reaction environment or in the ESI conditions. Regarding the mechanism of the formation of *DHmetformose*, it was hypothesized that the hydroxyl group implicated in this process was that linked to the quaternary carbon atom which reacts with the NH group(s) originally present in the metformin structure, i.e., implying different reacting sites and different structures of the reaction products. Then, it was considered that the water loss could lead, in principle, to different structures (A, B and C of Figure 3, originating via the reaction of the hydroxyl groups with NH groups in positions 2, 3 and 4, respectively).

### 2.3. HRMS and MS/MS Analysis to Characterize the Proposed Compounds

In order to confirm our hypotheses, HRMS measurements on the above-mentioned ion signals were carried out. The elemental composition was evaluated on the basis of each measured accurate *m*/*z* ratio, accepting only the results with an attribution error less than 2.5 ppm and a non-integer double bond/ring equivalent (RDB) value, so as to consider only the protonated species [31]. The HMRS results for the measured ions and their estimated elemental compositions are reported in Table 1.

The elemental compositions of the [M+H]^+^ ion species reported above for *metformose*, *DHmetformose* and metformin were consistent with the proposed structures (see Figure 2 and Figure 3), but any information about the functional groups and their arrangement were not gathered. To investigate the characteristics of the structures of these molecules, an ERMS study was performed, the collected MS/MS data were processed and the collision breakdown curves were plotted (see Figure 4 and Appendix A).

These graphs show that, after the activation of the fragmentation channels of each precursor ion, the product ions maintained a constant composition following the same trend despite the exitation amplitude (ExA) increasing. These patterns describe a typical collision induced dissociation (CID) behavior of MS/MS experiments performed in an ion trap (IT) mass analyzer [32,33]. Indeed, the breakdown curves demonstrate that all the fragmentation channels of the precursor ion in the IT were simultaneously activated at a defined ExA. Hence, after the complete dissociation of the precursor ion, the IT MS/MS spectra differed only in the yield of product ions formation, reaching their maximum value at a determined ExA (ExA_max_). The ExA_max_ obtained for each studied compound (see Table 2) was used as a specific ExA applied to monitor said compound in the MS/MS method developed to analyze the urine samples. The MS/MS spectra at the ExA_max_ values of *metformose*, *DHmetformose* and metformin are reported in Figure 5 and the Appendix A.

The MS/MS spectra of the studied compounds appeared very different from each other, except for the signal of the product ion at *m*/*z* 113, which was present in all three spectra. This product ion indicated a common structural moiety in the three compounds. Considering its origin from metformin, the simplest molecule, its formation was due to the loss of an ammonia molecule (−17 Da) from the [M+H]^+^ species. The other main ion fragments and the bond cleavages in metformin’s MS/MS spectrum are reported in the Appendix A. Following the same fragmentation pathway described above for the *metformose*-protonated molecule, the origins of the product ions at *m*/*z* 113 (base peak), *m*/*z* 222 and *m*/*z* 247 were assigned (Appendix A). The fragment at *m*/*z* 180 is interesting as it should be formed by the same bond cleavage of *m*/*z* 113 but with a different positive charge distribution. A separate discussion needs to be carried out in the interpretation of the MS/MS spectrum of *DHmetformose*. In fact, three different molecular structures have been proposed for this compound due to the different nitrogen involved in the cycle and the consequent loss of water. The product ions shown in the MS/MS spectrum confirm the model of the metformin fragmentation pathway, but they do not unequivocally clarify the structure of *DHmetformose*. Indeed, all its proposed isomers can produce these ions (Appendix A).

The collected data clearly demonstrate that the metformin reacted with glucose and that the reaction products were in agreement with the proposed structures of *metformose* and *DHmetformose*. However, these results cannot exclude the co-presence of isomers (e.g., structures A, B and C of *DHmetformose*); therefore, to clarify this point, further spectroscopic investigations are essential (^1^H- and ^13^C-NMR and FTIR), but some valid indication, even if only to be considered preliminary, was obtained by HRMS and MS/MS.

### 2.4. MS/MS Analysis of Urine Samples

The most important question in this study is: does the reaction between glucose and metformin occur in physiological conditions (temperature, pH, etc. …)? To investigate this aspect, urine samples from three untreated subjects (samples named B, C and D) (and one patient treated (sample A) with oral doses of metformin (1 + 1 g) were analyzed by 2D HPLC-MS/MS, monitoring the fragmentation products characteristic of metformin, *metformose* and *DHmetformose*. The application of the 2D HPLC technique was necessary to facilitate the analysis of the analytes in the human urine samples. In fact, it is known that human urine has high quantities of inorganic salts, which, if they coelute with the analytes in ESI conditions, can affect their ionization (matrix effects). Then, by injecting the sample into the loading column (first LC dimension), all the compounds that did not interact with the stationary phase were eliminated (e.g., inorganic salts), while the adsorbed analytes, after the loading time, were eluted in the counter-flow and analyzed from the analytical column (second LC dimension). The chromatographic profiles, related to the most abundant product ions in the MS/MS spectrum of each analyte, are reported in Figure 6.

As can be easily observed, in the case of the urine sample from the treated patient, the peak corresponding to metformin was present, but the peaks corresponding to *metformose* and *DHmetformose* were also detected. In the case of the urine samples from the controls, these three peaks were undetectable. It is worth noting that the retention times of *metformose* and *DHmetformose* were slightly different, proving that both these compounds were generated by the reaction of glucose with metformin and that *DHmetformose* is not an ESI-induced decomposition product of *metformose*. The nature of these peaks was confirmed by the MS/MS related spectra, showing the presence of the peaks characteristic of metformin, *metformose* and *DHmetformose* discussed above.

## 3. Materials and Methods

### 3.1. Chemicals

Acetonitrile, ethanol, methanol (Chromasolv grade), formic acid and ammonium formate (MS grade), acetic acid, metformin hydrochloride and D-(+)-glucose (reagent grade) were purchased from Merck (Milan, Italy). Ultrapure water or mQ water (resistivity 18 MΩ cm) was obtained from Millipore’s Simplicity system (Milan, Italy).

### 3.2. Synthesis of Glucose–Metformin Derivatives

The glucose–metformin derivatives were synthesized following a procedure reported in the literature [34] with slight modifications. Briefly, 2.20 g of metformin (17.0 mmol) and 3.07 g of D-glucose (17.0 mmol) were weighed in a pressure tube; after that, 30 mL of ethanol and 1 mL of acetic acid were added. The mixture was refluxed for 40 h and then cooled to room temperature. The solvent was removed under reduced pressure, and the residue was purified by flash chromatography on aluminum oxide (Al_2_O_3_) using 100% methanol as an eluent, yielding the desired compounds. The methanol was removed under reduced pressure, and 1 mg of the residue was diluted in a 1:1 ultrapure water:methanol (stock solution). The stock solution was maintained at 4 °C. Working solution 1 was prepared by diluting 0.1 mL of the stock solution up to 1 mL in the 1:1 ultrapure water:methanol mixture.

### 3.3. Instruments

The used high-performance liquid chromatography with a mass spectrometry detector (HPLC-MS) system was a Thermo LTQ Orbitrap XLhybrid mass spectrometer (Waltham, MA, USA) equipped with a Dionex Ultimate 3000 HPLC and an electrospray ion source (ESI). Raw data were collected and processed by XCalibur 2.0. An Eppendorf 5415D centrifuge (Merck, Milan, Italy) was employed to centrifuge the samples.

### 3.4. Two-Dimentional High-Performance Liquid Chromatography Method (2D-HPLC)

The chromatographic system employed for the analysis of the samples involved the use of three pumps and a six-port valve to perform the 2D-HPLC technique. The system was assembled around the six-port valve, which switched from the first to the second separation dimension. This technique led to the injection of the sample, in isocratic conditions, into the loading column (first dimension) via the loading pump; then, after swapping the position of the valve, the retained sample components were eluted in a counter-flow manner by analytical pumps and transferred into the chromatographic column (second dimension). Following this, the loading column received the sample and held the components of interest while the others were eluted to waste. These characteristics recall the solid-phase extraction (SPE) procedure but with the advantage of being performed on-line between the autosampler and the chromatographic column. In the second chromatographic dimension, the chromatographic column was coupled with the mass spectrometer and led to the separation of the retained components through a gradient elution carried out by the analytical pump system. The 2D-HPLC procedure described above is depicted in Figure 7.

The solvents used in the 2D-HPLC method were as follows: ultrapure water (solvent A), methanol (solvent B) and a 9:1 mixture solution of ultrapure water:methanol (solvent C). All the solvents were added with 20 mM formic acid and 5 mM ammonium formate. Solvents A and B were managed by the analytical pumps (second LC dimension), while solvent C was managed by the loading pump (first LC dimension). The chromatographic program started with the injection of 20 µL of the sample solution that was transferred from the loading pump in the Discovery® HS F5 Supelguard™ (Merck Life Science S.r.l., Milan, Italy) 20 × 2.1 mm cartridge at 3 µm (loading column), which was maintained for three minutes (loading time) with isocratic elution with solvent C at 0.5 mL min^−1^. After the loading time, the six-port valve swapped its position, allowing the counter-flow elution of the retained compounds, transferring them in a Phenomenex Luna PFP (2) (Phenomenex, Bologna, Italy) 50 × 2.1 mm device at 3 µm (analytical column). The analytical column was eluted at 0.25 mL min^−1^ with a solvent gradient composition provided by the analytical pump system, allowing for the separation of the analytes (Figure 7). The time program of the elution gradient was as follows: solvent B was held at 10% for three minutes, increasing up to 90% for four minutes and then remaining for five minutes, after which the initial conditions were restored. The loading and analytical columns were maintained at temperature of 40 °C.

### 3.5. Mass Spectrometry Methods

The MS spectra were acquired in the positive ion mode by using an electrospray ionization source (ESI) with the following set up: 5 kV source voltage, 35 arbitrary units (a.u.) of sheath gas, 5 a.u. of auxiliary gas, an entrance capillary voltage of 40 V and a temperature of 280 °C and a 20 V tube lens. The low-resolution mass spectra (MS spectra) were acquired using the LTQ ion trap of the hybrid mass spectrometer system by monitoring the positive ions in the range between *m*/*z* 80 and 350. An Orbitrap mass analyzer, set at 60,000 units of power resolution (FWHM), acquired the high-resolution mass spectrometry (HRMS) analyses in the range from *m*/*z* 80 to 350. The fragmentation study of the [M+H]^+^ species of each studied analyte and the selection of the appropriate excitation amplitude (ExA) to set up the tandem mass spectrometry (MS/MS) method were carried out by energy-resolved mass spectrometry (ERMS) experiments. The ERMS experiments consisted of a series of product ion scan (MS/MS) analyses at ExA values varying in the range of 0–50 arbitrary units (a.u.) with a fixed excitation time (ExT) of 50 milliseconds (ms). The ERMS experiments were performed by introducing working solution 1 (see Section 3.2) via a syringe pump at 10 µL min^−1^; the [M+H]^+^ species of the studied analyte were selected (precursor ions), and the related MS/MS spectra were registered (q = 0.25). The obtained data were used to build the graphs that describe the energetic dimension of the collision-induced dissociation (CID) process for each precursor ion [32,33].

The data from the ERMS experiments were processed and used to define the parameters to be set in the 2D HPLC-MS/MS method. These parameters could be common for all the analytes, such as an isolation width of 3 *m*/*z*, 0.25 as the q value and an excitation time (ExT) value of 50 ms, or they could be specific for each MS/MS event. The specific MS/MS parameters used for each analyte are listed and reported in Table 2.

### 3.6. Urine Sample Preparation

Each urine sample was prepared by 1:20 dilution with a 1:1 ultrapure water:methanol mixture in a microcentrifuge tube. The obtained solution was centrifuged at 10,000 r.p.m for 5 min, and the surnatant was transferred in an autosampler vial. The final sample solutions were analyzed by the above-described 2D HPLC-MS/MS method. The study was conducted in accordance with the Declaration of Helsinki, and the protocol was approved by the local Ethics Committee (Comitato Etico Regionale per la Sperimentazione Clinica della Regione Toscana, Sezione AREA VASTA CENTRO) Institutional Review Board (CE: 11166_spe, 11 September 2018 and CE: 10443_oss, 14 February 2017).

## 4. Conclusions

The results of the present study show that metformin reacts with glucose following the Maillard reaction pattern, leading to the formation of adducts originating from the condensation of the two molecular species with losses of one or two water molecules. These species, called *metformose* and *DHmetformose*, were detected in urine samples from diabetic patients treated with metformin, and consequently they must be considered metabolites of the drug, which have never been identified before now. It is worth to emphasize the chemical nature of these compounds that exhibit a glucose-related substructure, which could reflect a possibly better transfer ability through cell membranes. In fact, it must be considered that glucose is transferred inside the cells by specific membrane transporter proteins (GLUT proteins). Many isoforms of glucose transporter proteins have been identified in the human genome, and each of them changes its conformation to allow the passage of glucose to the other side of the cell membrane [29,35,36,37]. In particular, the isoform GLUT2, expressed by renal tubular cells, liver cells and pancreatic beta cells, is a bidirectional transporter, allowing glucose to flow in both directions (Figure 2). 

The above considerations seem to indicate that *metformose* and *DHmetfomose*, due to the presence of the glucose substructure, can be transported at the cellular level by GLUT proteins, still maintaining metformin’s activity. In other words, not only can organic cation transporter (OCT) be invoked to explain the action of metformin in liver cells, as described in the introduction section, but GLUT proteins can also play a role by interacting with alternative substrates (*metformose* and/or *DHmetformose*). The latter possibility opens the way to new hypotheses about the biological targets of metformin, in agreement with the results reported by Jeong and Jusko [21] and Bailey and Puah [22]. 

Interesting new data on metformin were presented in this manuscript, but it should be considered a preliminary investigation. Further studies are now in progress to evaluate the action of *metformose* and *DHmetformose* at the cellular level, their pharmacokinetics and the structural identification of *DHmetformose(s)* by other spectroscopic investigations (^1^H- and ^13^C-NMR and FTIR).

## Data Availability

Data is contained within the article.

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
