# Peer review of "Mass Spectrometry Study about In Vitro and In Vivo Reaction between Metformin and Glucose: A Preliminary Investigation on Alternative Biological Behavior"

_ijms, 2023, doi:10.3390/ijms25010180_

Round 1

Reviewer 1 Report

Comments and Suggestions for Authors

The manuscript titled "Mass spectrometry study about in vitro and in vivo reaction between Metformin and glucose, preliminary investigation on an alternative biological behavior " presents a comprehensive study investigating the reaction between metformin and glucose, shedding light on potential metabolites not identified before. The experimental design, including in vitro and in vivo conditions, mass spectrometry analyses, and the proposed structures of metformose and DHmetformose, adds valuable insights to the field. The discussion on the potential biological targets of metformin, involving both organic cation transporter (OCT) and glucose transporter (GLUT) proteins, introduces intriguing possibilities and aligns with recent research trends. I have only some minor comments (no major novel experiments are asked for), which are however important to bring the manuscript to a well-deserved higher level and impact.

(1)    While the proposed structures are compelling, further structural validation through additional spectroscopic techniques like 1H- and 13C-NMR and FTIR could strengthen the manuscript.

(2)    Isomeric Considerations: Given the possibility of isomers, a more detailed exploration or acknowledgment of the potential existence of structures A, B, and C of DHmetformose would enhance the discussion.

(3)    Cellular Level Impact: The manuscript hints at the potential impact of metformose and DHmetformose at the cellular level. A more explicit discussion or suggestions for future studies investigating their pharmacokinetics and cellular activities would be beneficial.

Comments on the Quality of English Language

The overall language of the manuscript is quite clear and technically sound.

Reviewer 2 Report

Comments and Suggestions for Authors

Overview and general recommendation:

The paper investigates the reaction of metformin, a widely used glucose-lowering agent used for the treatment of diabetes, with glucose. Reaction products were identified by HPLC-(HR)MS and MS/MS analyses and some hypothesis on the reaction mechanism and product structures were proposed. The reaction was first carried out in vitro for product characterization in a simplified environment and a biological sample (urine) was then analyzed, providing a proof of concept of the in vivo reaction occurrence.

The experiments were well-performed and described. Appropriate comments and assumptions were also made. Interestingly, metabolites of metformin were identified and characterized for the first time and the manuscript can be valuable for the scientific community.

Therefore, I believe this work is suitable for publication after addressing a few minor points.

Comments:

1) Page 5, line 168: please, check the paragraph number; it should be 2.2. In addition, I would suggest revising the title of this paragraph replacing “working solution 1” with something more meaningful for the aim of the paper.

2) I recommend briefly describing the composition of working solution 1 the first time it is discussed (page 5, line 169).

3) I think “section 2.2” should be replaced with “section 3.2” at lines 170 and 179 (page 5) and “Table 1” with “Table 2” at line 231 (page 7). Please, check.

4) In Paragraph 2.4, it would be interesting to add some explanation about the choice of performing a 2D HPLC-MS/MS analysis of the biological samples, since the method is different from the one used for the previous analyses.

5)      I suggest the authors specify the type of chromatogram in Figure 6. Is it an extracted ion chromatogram?

6)      I think that the sentence at lines 328-332 (page 11) is not immediately comprehensible and I recommend a revision.

Comments on the Quality of English Language

Minor editing of English language required.
